# Extending Kolmogorov’s Axioms for a Generalized Probability Theory on Collections of Contexts

**DOI:** 10.3390/e24091285

**Published:** 2022-09-12

**Authors:** Karl Svozil

**Affiliations:** Institute for Theoretical Physics, TU Wien, Wiedner Hauptstrasse 8-10/136, 1040 Vienna, Austria; svozil@tuwien.ac.at

**Keywords:** value indefiniteness, Kolmogorov axioms of probability theory, Pitowsky’s logical indeterminacy principle, quantum mechanics, Gleason theorem, Kochen–Specker theorem, Born rule

## Abstract

Kolmogorov’s axioms of probability theory are extended to conditional probabilities among distinct (and sometimes intertwining) contexts. Formally, this amounts to row stochastic matrices whose entries characterize the conditional probability to find some observable (postselection) in one context, given an observable (preselection) in another context. As the respective probabilities need not (but, depending on the physical/model realization, can) be of the Born rule type, this generalizes approaches to quantum probabilities by Aufféves and Grangier, which in turn are inspired by Gleason’s theorem.

## 1. Kolmogorov-Type Conditional Probabilities among Distinct Contexts

A physical system or a mathematical entity may permit not only one “view” on it but may allow “many” such views. For the sake of an illustration, imagine a metaphor mentioned by Schrödinger ([1] p. 15 and 95): a single crystal cluster whose light, depending on the viewpoint, may appear very different; the Vedantic analogy of a *“many-faceted crystal which, while showing hundreds of little pictures of what is in reality a single existent object, does not really multiply that object. … A comparison used in Hinduism is of the many almost identical images which a many-faceted diamond makes of some one object such as the sun.”* Another example is the coordinatization or coding and encryption of a vector with respect to different bases, thereby in physical terms appearing as “coherent superpositions” (linear combinations) of the respective vectors of these bases. Still another example is the representation of an entity by isomorphic graphs.

This idea is grounded in epistemology and in issues related to the (empirical) cognition of ontology and might appear both trivial and sophistic at first glance. Nevertheless, it may be difficult to find means or formal models exhibiting multiple contextual views of one and the same entity. Many conceptualizations of such situations are motivated by quantum complementarity [2,3,4,5].

A “view” or (used synonymously) “frame” [6] or “context” will be in full generality and thus informally (by glancing at heuristics from quantum mechanics and partition logic) characterized as some domain or set of observables or properties that is

(i)*largest* or *maximal* in the sense that any extension yields redundancies,(ii)yet at the same time in the *finest resolution* in the sense that the respective observables or properties are “no composite” of “more elementary” ones,(iii)contains only *mutually exclusive* observables in the sense that one property or observation excludes another, different property or observation, while at the same time(iv)includes only *simultaneously measurable, compatible* observables or properties.

In what follows, I shall develop a conceptual framework for very general probabilities on such collections of contexts. This amounts to an extension of Kolomogorov probabilities which are defined in a single context to a multi-context situation. Scrutinized separately, every single context has “legit local” classical Kolomogorov probabilities. In addition to those local structures and measures, (intertwined) multi-context configurations and their probabilities have to be “joined”, “woven”, “meshed” or “stitched” together to result in consistent and coherent “global” multi-aspect views and probabilities.

In particular, one needs to cope with possible overlaps of contexts in common, intertwining, observables. Because two or more contexts need not (but may) be separated from one another, they may indeed intertwine in one or more common elements and form complex propositional structures serving a variety of counterfactual [7,8] purposes [9,10,11].

In terms of probabilistic language, one might interpret contexts as conditions. Intertwining contexts might be identified with different conditionings with non-empty intersection(s).

This text is organized as follows: first, Kolmogorov’s axioms or principles for probabilities are generalized to arbitrary event structures not necessarily dominated by the quantum formalism. Then these principles will be applied to quantum bistochasticity, as well as partition logics which offer an abundance of alternate configurations. Some “exotic” probabilities as well as possible generalizations by Cauchy’s functional equations are briefly discussed. Throughout this article, only finite contexts will be considered.

## 2. Generalization of Kolmogorov’s Axioms to Arbitrary Event Structures

Suppose that, as it is assumed for classical Kolmogorov probabilities, the elements $c_{1}$ within any given single, individual finite context $C=\{c_{1},\ldots c_{n}\}$ are mutually exclusive, compatible and exhaustive; that is, the context contains a “maximal” set of mutually exclusive compatible elements. Kolmogorov’s axioms demand that (i) probabilities are non-negative; (ii) additivity of mutually exclusive events or outcomes $P\left(c_{i}\right)+P\left(c_{j}\right)=P\left(c_{i}\cup c_{j}\right)$; (iii) the probability of the tautology formed by the union of all elements in the context adds up to one; that is, $\sum_{c_{i}\in C}P\left(c_{i}\right)=P\left({⋃}_{c_{i}\in C}c_{i}\right)=1$.

Inspired by the multi-context quantum case discussed later, the following generalization to two- or, by induction, to a multi-context configuration is suggested: Suppose two arbitrary contexts $C_{1}=\left\{e_{1},\ldots e_{n}\right\}$ and $C_{2}=\left\{f_{1},\ldots f_{m}\right\}$. The conditional probabilities $P(f_{j}|e_{i})$, with 1≤j≤m and 1≤i≤n, which alternatively can be considered as either measuring the Bayesian degree of reasonable expectation representing a state of knowledge or as quantification of a personal belief [12] or the frequency of occurrence of “$f_{j}$ given $e_{i}$”, can be arranged into a (n×m)-matrix whose entries are $P(f_{j}|e_{i})$,=; that is,
(1)$$\begin{array}{c} \left[P(C_{2}|C_{1})\right]=\left[P(\left\{f_{1},\ldots f_{m}\right\}|\left\{e_{1},\ldots e_{n}\right\})\right]\\ \equiv \left[\begin{array}{ccc} P(f_{1}|e_{1}) & \cdots & P(f_{m}|e_{1})\\ \cdots & \cdots & \cdots \\ P(f_{1}|e_{n}) & \cdots & P(f_{m}|e_{n}) \end{array}\right]. \end{array}$$

Assume as axiom the following criterion: the conditional probabilities of the elements of the second context with respect to an arbitrary element $e_{k}\in C_{1}$ of the first context $C_{1}$ are non-negative, additive, and, if this sum is extended over the entire second context $C_{2}$, add up to one:(2)$$\begin{array}{c} P\left(f_{i}|e_{k}\right)+P\left(f_{j}|e_{k}\right)=P\left[\left(f_{i}\cup f_{j}\right)|e_{k}\right]\\ \sum_{f_{i}\in C_{2}}P\left(f_{i}|e_{k}\right)=P\left[\left(\underset{f_{i}\in C_{2}}{⋃}f_{i}\right)|e_{k}\right]=1. \end{array}$$
That is, the row sum taken within every single row of $\left[P(C_{2}|C_{1})\right]$ adds up to one.

This presents a generalization of Kolmogorov’s axioms, as it allows cases in which both contexts do not coincide. It just reduces to the classical axioms for single contexts if, instead of a single element $e_{k}\in C_{1}$ of the first context $C_{1}$, the union of elements of this entire context $C_{1}$—and thus the tautology ${⋃}_{e_{i}\in C_{1}}e_{i}$—is inserted into (Equation 2).

We shall mostly be concerned with cases for which n=m; that is, the associated matrix is a row (right) stochastic (square) matrix. Formally, such a matrix A has nonnegative entries $a_{ij}\geq 0$ for i,j=1,…,n whose row sums add up to one: $\sum_{j=1}^{n}a_{ij}=1$ for i=1,…,n. If, in addition to the row sums, also the column sums add up to one; that is, if $\sum_{i=1}^{n}a_{ij}=1$ for j=1,…,n, then the matrix is called doubly stochastic. If J is a (n×n)–matrix whose entries are 1, then a (n×n)–matrix A is row stochastic if AJ=J.

It is instructive to ponder why intuitively those conditional probabilities should be arranged in right- but not in bistochastic matrices. Suppose a (physical or another model) system is in a state characterized by some element $e_{j}\in C_{1}$ of the first context $C_{1}$. Then, if one takes the (union of elements of the) entire other context $C_{2}$, thereby exhausting all possible outcomes of the second “view”, the conditional probability for this system to be in *any* element of $C_{2}$ given $e_{j}\in C_{1}$ should add up to one because this includes all that can be (or happen or exist) with respect to the second “view”. Indeed, if this conditional probability would not add up to one, say if it adds up to something strictly smaller or larger than one, then either some elements would be missing in, or be “external” to, the context $C_{2}$, which cannot occur since by assumption contexts are “maximal”.

On the other hand, if a particular element $f_{i}\in C_{2}$ of the second context $C_{2}$ remains fixed, and the column sum $\sum_{e_{j}\in C_{1}}P\left(f_{i}|e_{j}\right)$ extends over all $e_{j}\in C_{1}$, then there is no convincing reason why this column sum should add up to one. Indeed, as will be argued later, while quantum mechanics results in bistochastic matrices for pure states, generalized urn models result in partitions of (hidden) variables that will not induce bistochasticity.

## 3. Cauchy’s Functional Equation Encoding Additivity

One way of looking at generalized global probabilities from “stitching” local classical Kolmogorov probabilities is to maintain the essence of the axioms, namely positivity, probability one (certainty) for tautologies and, in particular, additivity. Additivity requires that, for mutually exclusive compatible events $c_{i}$ and $c_{j}$ within a given context, their probabilities can be expressed in terms of a Cauchy-type functional equation $P\left(c_{i}\right)+P\left(c_{j}\right)=P\left(c_{i}\cup c_{j}\right)$. With “reasonable” side assumptions, this amounts to the linearity of probabilities in the argument [13,14].

For operators in Hilbert spaces of dimensions higher than two, and in particular for linear operators A and B with an operator norm $\left|A\right|=+\sqrt{\langle A|A\rangle }$ based on the Hilbert–Schmidt inner product $\langle A|B\rangle =\mathrm{Trace}\left(A^{*}B\right)$, where $A^{*}$ stands for the adjoint of A, Cauchy’s functional equation can be related to Gleason-type theorems [15,16,17,18,19,20].

The general case may involve other, hitherto unknown, arguments besides scalars and entities related to vector (or Hilbert) spaces. The discussion will not be extended to potential inputs and sources for generalized probabilities as the main interest is in developing a generalizing probability theory in the multi-context setting, but clearly these questions remain pertinent.

## 4. Examples of Application of the Generalized Kolmogorov Axioms

### 4.1. Quantum Bistochasticity for Pure States

The multi-context quantum case has been studied in great detail with emphasis on motivating and deriving the Born rule [21,22] from elementary foundations. Recall that a context has been defined as the “largest” or “maximal” domain of both mutually exclusive as well as simultaneously measurable, compatible observables. In quantum mechanics, “simultaneously measurability” transforms into *compatibility* and *commutativity*; that is, such observables are not complementary and can be jointly measured without restrictions. “Mutual exclusivity” is defined in terms of *orthogonality* of the respective observables. The spectral theorem asserts mutual orthogonality of unit eigenvectors $|e_{i}\rangle$ and the associated orthogonal projection operators $E_{i}$ formed by the dyadic product $E_{i}=|e_{i}\rangle \langle e_{i}|$. A context can be equivalently represented by (i) an orthonormal basis, (ii) the respective one-dimensional orthogonal projection operators associated with the basis elements, or (iii) a single maximal operator (maximal observable) whose spectral sum is non-degenerate [9,23].

An essential assumption entering Gleason’s derivation [6] of the Born rule for quantum probabilities is the validity of classical probability theory whenever the respective observables are compatible. Formally, this amounts to the validity of Kolmogorov probability theory for mutually commuting observables; and in particular, to the assumption of Kolmogorov’s axioms within contexts.

Already Gleason pointed out [6] that it is quite straightforward to find an ad hoc probability satisfying this aforementioned assumption, which is based on the Pythagorean property: suppose (i) a quantized system is in a pure state |ψ〉 formalized by some unit vector, and (ii) some “measurement frame” formalized by an orthonormal basis $C=\{|e_{1}\rangle ,\ldots ,|e_{n}\rangle \}$. Then, the probabilities of outcomes of observable propositions associated with the orthogonal projection operators formed by the dyadic products $|e_{i}\rangle \langle e_{i}|$ of the vectors of the orthonormal basis can be obtained by taking the absolute square of the length of those projections of |ψ〉 onto $|e_{i}\rangle$ along the remaining basis vectors, which amounts to taking the scalar products $|\langle \psi |e_{i}\rangle {|}^{2}$. Since the vector associated with the pure state as well as all the vectors in the orthonormal system are of length one, and since these latter vectors (of the orthonormal system) are mutually orthogonal, the sum $\sum_{i=1}^{n}{\left|\langle \psi |e_{i}\rangle \right|}^{2}$ of all these terms, taken over all the basis elements, needs to add up to one. The respective absolute squares are bounded between zero and one. In effect, the orthonormal basis “grants a view” of the pure quantum state. The absolute square can be rewritten in terms of a trace (over some arbitrary orthonormal basis) into the standard form known as the Born rule of quantum probabilities: $|\langle \psi |e_{i}\rangle {|}^{2}=\langle \psi |e_{i}\rangle \langle e_{i}|\psi \rangle =\langle \psi |e_{i}\rangle \langle e_{i}|I_{n}\psi \rangle =\sum_{j=1}^{n}\langle \psi |e_{i}\rangle \langle e_{i}|g_{j}\rangle \langle g_{j}|\psi \rangle =\sum_{j=1}^{n}\langle g_{j}|\underset{=E_{\psi }}{\underbrace{\psi \rangle \langle \psi }}|\underset{=E_{i}}{\underbrace{e_{i}\rangle \langle e_{i}}}|g_{j}\rangle =\mathrm{Trace}\left(E_{\psi }E_{i}\right)$, where $E_{\psi }$ and $E_{i}$ are the orthogonal projection operators representing the state |ψ〉 and the (unit) vectors of the orthonormal basis $|e_{i}\rangle$, respectively, and $C^{\prime }=\{|g_{1}\rangle ,\ldots ,|g_{n}\rangle \}$ is an arbitrary orthonormal basis, so that a resolution of the identity is $I_{n}=\sum_{j=1}^{n}|g_{j}\rangle \langle g_{j}|$.

It is also well known that, at least from a formal perspective, unit vectors in quantum mechanics serve a dual role. On the one hand, they represent pure states. On the other hand, by the associated one-dimensional orthogonal projection operator, they represent an observable: the proposition that the system is in such a pure state [24,25]. Suppose now that we exploit this dual role by *expanding* the pure prepared state into a full orthonormal basis, of which its vector must be an element. (For dimensions greater than two, such an expansion will not be unique as there is a continuous infinity of ways to achieve this.) Once the latter basis is fixed, it can be used to obtain a “view” on the former (measurement) basis, and a completely symmetric situation/configuration is attained. We might even go so far as to say that which basis is associated with the “observed object” and with the “measurement apparatus”, respectively, is purely a matter of convention and subjective perspective.

Therefore, as has been pointed out earlier, an orthogonal projection operator serves a dual role. On the one handm it is a formalization of a dichotomic observable, more precisely, an elementary yes–no proposition E=|x〉〈x| associated with the claim that “the quantized system is in state |x〉. On the other hand, it is the formal representation of a pure quantum state |y〉 equivalent to the operator F=|y〉〈y|. By the Born rule, the conditional probabilities are symmetric with respect to exchange of |x〉 and |y〉: let $C^{\prime }=\{|g_{1}\rangle ,\ldots ,|g_{n}\rangle \}$ be some arbitrary orthonormal basis of $C^{n}$, then $P\left(E|F\right)=\mathrm{Trace}\left(EF\right)=\mathrm{Trace}\left(FE\right)=P\left(F|E\right)$; or, more explicitly, $P\left(E|F\right)=\sum_{i=1}^{n}\langle g_{i}|x\rangle \langle x|y\rangle \langle y|g_{i}\rangle =\sum_{i=1}^{n}\langle x|y\rangle \langle y|\underset{=I_{n}}{\underbrace{g_{i}\rangle \langle g_{i}}}|x\rangle ={\left|\langle x|y\rangle \right|}^{2}={\left|\langle y|x\rangle \right|}^{2}=P\left(F|E\right)$. Therefore, the respective conditional probabilities form a doubly stochastic (bistochastic) square matrix. This result is a special case of a more general result on quadratic forms on the set of eigenvectors of normal operators [26].

Consider two orthonormal bases (two contexts). Their respective conditional probabilities can be arranged into a matrix form. The *i*th row *j*th column component corresponds to the conditional probability associated with the probability of occurrence of the *j*th element (observable) of the second context, given the *i*th element (observable) of the first context. By taking into account that cyclically interchanging factors inside a trace does not change its value, this matrix needs to be not only row (right) stochastic but doubly stochastic (bistochastic) [21,22]; that is, the sum is taken within every single row and every single column adds up to one.

It is important to emphasize that bistochasticity holds for pure states but not for more general ones. In particular, for non-rank-one density matrices that are the product of two vectors, such as for mixed states, the above arguments do not apply.

### 4.2. Quasi-Classical Partition Logics

In what follows, we shall study sets of partitions of a given set. They have models [27] based on (i) the finite automata initial state identification problem [28] as well as (ii) generalized urns [29,30]. Partition logics are quasi-classical and value-definite in so far as they allow a separating set of “classical” two-valued states [9] (Theorem 0), and yet they feature complementarity. Many of these logics are *doubles* of quantum logics, such as for spin-state measurements, and thereby their graphs also allow faithful orthogonal representations [31]. Yet some of them have no quantum analog. Therefore, they neither form a proper subset of all quantum logics nor do they contain all logical structures encountered in quantum logics (they are neither continuous nor can they have a non-separating or nonexisting set of two-valued states). However, partition logics overlaps significantly with quantum logics, as they bear strong similarities with the structures arising in quantum theory.

If some (partition) logic which is a pasting [32,33,34] of contexts has a separating set of two-valued states [9] (Theorem 0), then there is a constructive, algorithmic [35] way of finding a “canonical” partition logic [27], and associated with it, all classical probabilities on it. First, find all the two-valued states on the logic and assign consecutive number to these states. Then, for any atom (element of a context), find the index set of all two-valued states which are one on this atom. Associate with each one, say, the *i*th, of the two valued states a nonnegative weight $i\to \lambda_{i}$, and require that the (convex) sum of these weights $\sum_{i}\lambda_{i}=1$ is 1. Since all two-valued states are included, the Kolmogorov axioms guarantee that the sum of measures/weights within each of the contexts in the logic exactly adds up to one.

It will be argued that in this case, and unlike for quantum conditional probabilities, the conditional probabilities, in general, do not form a bistochastic matrix.

#### 4.2.1. Two Non-Intertwining Two-Atomic Contexts

In the Babylonian spirit ([36] p. 172), consider some anecdotal examples which have quantum doubles. The first one will be analogous to a spin-$\frac{1}{2}$ state measurement.

The logic in Figure 1 enumerates the labels of the atoms (elementary propositions) according to the “inverse construction” based on all four two-valued states on the logic mentioned earlier, using all two-valued measures thereon [27]. With the identifications $e_{1}\equiv \left\{1,2\right\}$, $e_{2}\equiv \left\{3,4\right\}$, $f_{1}\equiv \left\{1,3\right\}$, and $f_{2}\equiv \left\{2,4\right\}$, we obtain all classical probabilities by identifying $i\to \lambda_{i}>0$. The respective conditional probabilities are
(3)$$\begin{array}{c} \left[P(C_{2}|C_{1})\right]=\left[P(\left\{f_{1},f_{2}\right\}|\left\{e_{1},e_{2}\right)\right]\equiv \left[\begin{array}{cc} P(f_{1}|e_{1}) & P(f_{2}|e_{1})\\ P(f_{1}|e_{2}) & P(f_{2}|e_{2}) \end{array}\right]\\ =\left[\begin{array}{cc} \frac{P(f_{1}\cap e_{1})}{P(e_{1})} & \frac{P(f_{2}\cap e_{1})}{P(e_{1})}\\ \frac{P(f_{1}\cap e_{2})}{P(e_{2})} & \frac{P(f_{2}\cap e_{2})}{P(e_{2})} \end{array}\right]=\left[\begin{array}{cc} \frac{P(\{1,3\}\cap \{1,2\})}{P(\{1,2\})} & \frac{P(\{2,4\}\cap \{1,2\})}{P(\{1,2\})}\\ \frac{P(\{1,3\}\cap \{3,4\})}{P(\{3,4\})} & \frac{P(\{2,4\}\cap \{3,4\})}{P(\{3,4\})} \end{array}\right]\\ =\left[\begin{array}{cc} \frac{P(\{1\})}{P(\{1,2\})} & \frac{P(\{2\})}{P(\{1,2\})}\\ \frac{P(\{3\})}{P(\{3,4\})} & \frac{P(\{4\})}{P(\{3,4\})} \end{array}\right]=\left[\begin{array}{cc} \frac{\lambda_{1}}{\lambda_{1}+\lambda_{2}} & \frac{\lambda_{2}}{\lambda_{1}+\lambda_{2}}\\ \frac{\lambda_{3}}{\lambda_{3}+\lambda_{4}} & \frac{\lambda_{4}}{\lambda_{3}+\lambda_{4}} \end{array}\right], \end{array}$$
as well as
(4)$$\begin{array}{c} \left[P(C_{1}|C_{2})\right]=\left[P(\left\{e_{1},e_{2}\right\}|\left\{f_{1},f_{2}\right\})\right]\\ \equiv \left[\begin{array}{cc} \frac{P(\{1\})}{P(\{1,3\})} & \frac{P(\{3\})}{P(\{1,3\})}\\ \frac{P(\{2\})}{P(\{2,4\})} & \frac{P(\{4\})}{P(\{2,4\})} \end{array}\right]=\left[\begin{array}{cc} \frac{\lambda_{1}}{\lambda_{1}+\lambda_{3}} & \frac{\lambda_{3}}{\lambda_{1}+\lambda_{3}}\\ \frac{\lambda_{2}}{\lambda_{2}+\lambda_{4}} & \frac{\lambda_{4}}{\lambda_{2}+\lambda_{4}} \end{array}\right]. \end{array}$$

#### 4.2.2. Two Intertwining Three-Atomic Contexts

In what follows, we shall investigate a “firefly” model that has been introduced ([38] Figure 3A.1, p. 22) to investigate a quasi-classical example of an empirical situation occurring in quantized systems with three exclusive outcomes formalized by three-dimensional Hilbert space. It comprises a box with two perpendicular windows and a firefly inside. Suppose that sometimes the firefly radiates some light, and sometimes it does not shine. Suppose further that each one of the two perpendicular windows has a thin vertical line drawn down the center to divide the respective window in half.

This configuration allows two types of experiments corresponding to looking through exactly one of the two windows, respectively. Each type of experiment has three outcomes, labeled as follows:(i)$e_{1}$ (first type of experiment) or $f_{1}$ (second type of experiment): the light of the firefly is in the left half of the window;(ii)$e_{2}$ (first type of experiment) or $f_{2}$ (second type of experiment): the light of the firefly is in the right half of the window;(iii)$e_{3}$ (first type of experiment) and $f_{3}$ (second type of experiment): the firefly does not shine (does not emit light).

The two observers at the two windows may observe any of the four combinations $e_{1}$ or $f_{1}$, $e_{1}$ or $f_{2}$, $e_{2}$ or $f_{1}$, or $e_{2}$ or $f_{2}$. Ideally, it will always be the case that whenever the first observer registers no light—that is, $e_{3}$—also the second observer will register no light—that is, $f_{3}$—and vice versa.

This firefly configuration thus gives rise to two contexts $\left\{e_{1},e_{2},e_{3}\right\}$ and $\left\{f_{1},f_{2},f_{3}\right\}$, associated with the two observers, respectively. These contexts are “tied together” and intertwine at the “no light” event or outcomes $e_{3}$ and $f_{3}$. Together, this results in five conceivable experimental outcomes for two observers, corresponding to five two-valued measures representing these outcomes, respectively.

The associated $L_{12}$ firefly logic depicted in Figure 2 uses labels for the atoms (elementary propositions) that can be obtained by an “inverse construction” using all its five two-valued measures [27,39]. By design, it will be very similar to the earlier logic with four atoms. With the identifications $e_{1}\equiv \left\{1,2\right\}$, $e_{2}\equiv \left\{3,4\right\}$, $e_{3}=f_{3}\equiv \left\{5\right\}$, $f_{1}\equiv \left\{1,3\right\}$ and $f_{2}\equiv \left\{2,4\right\}$, we obtain all classical probabilities by identifying $i\to \lambda_{i}>0$. The respective conditional probabilities are
(5)$$\begin{array}{c} \left[P(C_{2}|C_{1})\right]=\left[P(\left\{f_{1},f_{2},f_{3}\right\}|\left\{e_{1},e_{2},e_{3}\right\})\right]\\ \equiv \left[\begin{array}{ccc} \frac{P(\{1\})}{P(\{1,2\})} & \frac{P(\{2\})}{P(\{1,2\})} & \frac{P(\emptyset )}{P(\{1,2\})}\\ \frac{P(\{3\})}{P(\{3,4\})} & \frac{P(\{4\})}{P(\{3,4\})} & \frac{P(\emptyset )}{P(\{3,4\})}\\ \frac{P(\emptyset )}{P(\{5\})} & \frac{P(\emptyset )}{P(\{5\})} & \frac{P(\{5\})}{P(\{5\})} \end{array}\right]=\left[\begin{array}{ccc} \frac{\lambda_{1}}{\lambda_{1}+\lambda_{2}} & \frac{\lambda_{2}}{\lambda_{1}+\lambda_{2}} & 0\\ \frac{\lambda_{3}}{\lambda_{3}+\lambda_{4}} & \frac{\lambda_{4}}{\lambda_{3}+\lambda_{4}} & 0\\ 0 & 0 & 1 \end{array}\right], \end{array}$$
as well as
(6)$$\begin{array}{c} \left[P(C_{1}|C_{2})\right]=\left[P(\left\{e_{1},e_{2},e_{3}\right\}|\left\{f_{1},f_{2},f_{3}\right\})\right]\\ \equiv \left[\begin{array}{ccc} \frac{P(\{1\})}{P(\{1,3\})} & \frac{P(\{3\})}{P(\{1,3\})} & \frac{P(\emptyset )}{P(\{1,3\})}\\ \frac{P(\{2\})}{P(\{2,4\})} & \frac{P(\{4\})}{P(\{2,4\})} & \frac{P(\emptyset )}{P(\{2,4\})}\\ \frac{P(\emptyset )}{P(\{5\})} & \frac{P(\emptyset )}{P(\{5\})} & \frac{P(\{5\})}{P(\{5\})} \end{array}\right]=\left[\begin{array}{ccc} \frac{\lambda_{1}}{\lambda_{1}+\lambda_{3}} & \frac{\lambda_{3}}{\lambda_{1}+\lambda_{3}} & 0\\ \frac{\lambda_{2}}{\lambda_{2}+\lambda_{4}} & \frac{\lambda_{4}}{\lambda_{2}+\lambda_{4}} & 0\\ 0 & 0 & 1 \end{array}\right]. \end{array}$$

The conditional probabilities of the firefly logic, as depicted in Figure 2a, and enumerated in Equation (Equation 6) form a right stochastic matrix. As mentioned earlier, given any particular outcome $f_{i}$ of the second context corresponding to some respective row in the matrix (Equation 6), the row-sum of the conditional probabilities of all the conceivable mutually exclusive outcomes of the first context $C_{1}=\left\{e_{1},e_{2},e_{3}\right\}$ must be one. However, the “transposed” statement is not true. The column-sum of the conditional probabilities of a particular element $e_{j}$ with respect to all the mutually exclusive outcomes of the second context $C_{2}=\left\{f_{1},f_{2},f_{3}\right\}$ needs not be one.

Take, for example, the singular distribution case such that $\lambda_{1}=1$, and therefore, by positivity and convexity, $\lambda_{i\neq 1}=0$; that is, $\lambda_{2}=\lambda_{3}=\lambda_{4}=\lambda_{5}=0$. This configuration, depicted in Figure 2c, results in the following, partial (undefined components are indicated by the symbol “$\frac{0}{0}$”) right stochastic matrix (Equation 7) derived from (Equation 6):(7)$$\begin{array}{c} \left[\begin{array}{ccc} \frac{P(\{1\})}{P(\{1,3\})} & \frac{P(\{3\})}{P(\{1,3\})} & \frac{P(\emptyset )}{P(\{1,3\})}\\ \frac{P(\{2\})}{P(\{2,4\})} & \frac{P(\{4\})}{P(\{2,4\})} & \frac{P(\emptyset )}{P(\{2,4\})}\\ \frac{P(\emptyset )}{P(\{5\})} & \frac{P(\emptyset )}{P(\{5\})} & \frac{P(\{5\})}{P(\{5\})} \end{array}\right]\\ =\left[\begin{array}{ccc} \frac{\lambda_{1}}{\lambda_{1}+\lambda_{3}} & \frac{\lambda_{3}}{\lambda_{1}+\lambda_{3}} & 0\\ \frac{\lambda_{2}}{\lambda_{2}+\lambda_{4}} & \frac{\lambda_{4}}{\lambda_{2}+\lambda_{4}} & 0\\ 0 & 0 & 1 \end{array}\right]=\left[\begin{array}{ccc} 1 & 0 & 0\\ \frac{0}{0} & \frac{0}{0} & 0\\ 0 & 0 & 1 \end{array}\right]. \end{array}$$
In such a case, in terms of, say, a generalized urn model, the observable proposition {2,4} associated with the plaintext *“looked upon in the first color (in this case blue), the ball drawn from the urn shows the symbols 2 or 4”* will never occur; regardless of which ball type associated with the other context {1,2}, {3,4} or {5} one would have (counterfactually) drawn because the generalized urn is only loaded with balls of one type, namely the first type, with the symbol “{1,2}” painted on them in the first color, and the symbols “{1,3}” painted on them in the second color. (Instead of labels indicating the elements of the partition, one may choose other symbols, such as {1,3}≡a≡{1,2}, {2,4}≡b≡{3,4} and c≡{5}, in the respective colors [27,40]).

Ultimately, one may say that it is the *discontinuity* of the two-valued measures which “prevents” the quasiclassical conditional probabilities to be arranged in a bistochastic matrix. A similar quantum realization could, for instance, be obtained by the three-dimensional faithful orthogonal representation [37] $\left\{1,2\right\}\equiv {\left(\begin{array}{c} 1,0,0 \end{array}\right)}^{⊺}$, $\left\{3,4\right\}\equiv {\left(\begin{array}{c} 0,1,0 \end{array}\right)}^{⊺}$, $\left\{5\right\}\equiv {\left(\begin{array}{c} 0,0,1 \end{array}\right)}^{⊺}$, $\left\{1,3\right\}\equiv \left(1/\sqrt{2}\right){\left(\begin{array}{c} 1,1,0 \end{array}\right)}^{⊺}$ and $\left\{2,4\right\}\equiv \left(1/\sqrt{2}\right){\left(\begin{array}{c} 1,-1,0 \end{array}\right)}^{⊺}$. Preparition (“loading the quantum urn”) with state $\left\{1,2\right\}\equiv {\left(\begin{array}{c} 1,0,0 \end{array}\right)}^{⊺}$, as depicted in Figure 2d, yields the quantum bistochastic matrix
(8)$$\begin{array}{c} \left[P\left(\left\{\left(\begin{array}{c} 1\\ 0\\ 0 \end{array}\right),\left(\begin{array}{c} 0\\ 1\\ 0 \end{array}\right),\left(\begin{array}{c} 0\\ 0\\ 1 \end{array}\right)\right\}|\left\{\left(\begin{array}{c} \frac{1}{\sqrt{2}}\\ \frac{1}{\sqrt{2}}\\ 0 \end{array}\right),\left(\begin{array}{c} \frac{1}{\sqrt{2}}\\ -\frac{1}{\sqrt{2}}\\ 0 \end{array}\right),\left(\begin{array}{c} 0\\ 0\\ 1 \end{array}\right)\right\}\right)\right]=\left[\begin{array}{ccc} \frac{1}{2} & \frac{1}{2} & 0\\ \frac{1}{2} & \frac{1}{2} & 0\\ 0 & 0 & 1 \end{array}\right]. \end{array}$$

#### 4.2.3. Different Intrinsically Operational State Preparation

A different approach to partition logic would be to insist that only *intrinsical*—that is, for any embedded observer having access to means and methods available “from within” the system—operational state preparations should be allowed. In such a scenario, it is operationally impossible for an observer with access to only one context—in the generalized urn model only one color—to single out the particular type of two-valued measure (ball). Thereby, effectively any state preparation is reduced to the elements of the partition in the respective context (color).

Therefore, in the earlier firefly model depicted in Figure 2, the intrinsic operational resolution is among the *subsets resulting from the unions of two-valued states* in {1,2}, {3,4} and {5} in the first context (color); and among {1,3}, {2,4} and {5} in the second context ( color), as opposed to the single two-valued state discussed earlier. Stated differently, an observer accessing a generalized urn in the first context (color) is not capable to differentiate between the first and the second two-valued measure (ball type), and would produce a mixture among them if asked to prepare the state {1,2}. Similarly, the observer would not be able to differentiate between the third and the fourth two-valued measure (ball type) and would thus produce a mixture between those when preparing the state {3,4}. However, the ball type {5} is recognized and prepared without ambiguity. Indeed, if one assumes equidistribution (uniform mixtures ([41] Assumption 1)) of measures (ball types), a very similar situation as in quantum mechanics [cf Figure 2d, Equation (Equation 8)] would result as $\lambda_{1}=\lambda_{2}=\lambda_{3}=\lambda_{4}=\lambda_{5}=\frac{1}{5}$, and one would thus “recover” the matrix in Equation (Equation 8).

Pointedly stated, there is an epistemic issue of state preparation. If one demands that the state has to be prepared by the distinctions accessible from a single context (color in the generalized urn model), then there is no way to prepare or access “ontologic states”, say, selecting balls of type 1 (first two-valued measure) only. The difference is subtle. In the “ontic” state case, one can resolve (and has access to) every single two-valued measure (ball type). In the “epistemic”, intrinsic, operational state case, one is limited to the operational procedures available. For example, one cannot “take off the colored glasses” in Wright’s generalized urn model. That is, the resolution of balls is limited to whatever types can be differentiated in that color.

Whenever such a scenario is considered, the respective matrices representing all conditional probabilities may be very different from the previous scenarios. Indeed, one may suspect that with the assumption of preservation of equidistributed uniform mixtures across context changes, the respective matrices are bistochastic (at least for equidistributed urns) because of a certain type of “epistemic continuity”. The sum of the conditional probabilities for any particular outcome of the second context relative to all other outcomes of the first context should add up to unity.

#### 4.2.4. Pentagon/Pentagram/House Logic with Five Cyclically Intertwining Three-Atomic Contexts

By now, it should be clear how classical conditional probabilities work on partition logics. Consider one more example: the pentagon/pentagram/(orthomodular) house ([33] [p. 46, Figure 4.4]) logic in Figure 3. Labels of the atoms (elementary propositions) are again obtained by an “inverse construction” using all 11 two-valued measures thereon [29]. Take, for example, one of the two contexts $C_{4}=\left\{\left\{2,7,8\right\},\left\{1,3,9,10,11\right\},\left\{4,5,6\right\}\right\}$ “opposite” to the context $C_{1}=\left\{\left\{1,2,3\right\},\left\{4,5,7,9,11\right\},\left\{6,8,10\right\}\right\}$.

With the identifications $e_{1}\equiv \left\{1,2,3\right\}$, $e_{2}\equiv \left\{4,5,7,9,11\right\}$, $e_{3}\equiv \left\{6,8,10\right\}$, $f_{1}\equiv \left\{2,7,8\right\}$, $f_{2}\equiv \left\{1,3,9,10,11\right\}$, and $f_{3}\equiv \left\{4,5,6\right\}$. The respective conditional probabilities are
(9)$$\begin{array}{c} \left[P(C_{2}|C_{1})\right]=\left[P(\left\{f_{1},f_{2},f_{3}\right\}|\left\{e_{1},e_{2},e_{3}\right\})\right]\\ \equiv \left[\begin{array}{ccc} \frac{P(\{2,7,8\}\cap \{1,2,3\})}{P(\{1,2,3\})} & \frac{P(\{1,3,9,10,11\}\cap \{1,2,3\})}{P(\{1,2,3\})} & \frac{P(\{4,5,6\}\cap \{1,2,3\})}{P(\{1,2,3\})}\\ \frac{P(\{2,7,8\}\cap \{4,5,7,9,11\})}{P(\{4,5,7,9,11\})} & \frac{P(\{1,3,9,10,11\}\cap \{4,5,7,9,11\})}{P(\{4,5,7,9,11\})} & \frac{P(\{4,5,6\}\cap \{4,5,7,9,11\})}{P(\{4,5,7,9,11\})}\\ \frac{P(\{2,7,8\}\cap \{6,8,10\})}{P(\{6,8,10\})} & \frac{P(\{1,3,9,10,11\}\cap \{6,8,10\})}{P(\{6,8,10\})} & \frac{P(\{4,5,6\}\cap \{6,8,10\})}{P(\{6,8,10\})} \end{array}\right]\\ =\left[\begin{array}{ccc} \frac{P(\{2\})}{P(\{1,2,3\})} & \frac{P(\{1,3\})}{P(\{1,2,3\})} & \frac{P(\emptyset )}{P(\{1,2,3\})}\\ \frac{P(\{7\})}{P(\{4,5,7,9,11\})} & \frac{P(\{11\})}{P(\{4,5,7,9,11\})} & \frac{P(\{4,5\})}{P(\{4,5,7,9,11\})}\\ \frac{P(\{8\})}{P(\{6,8,10\})} & \frac{P(\{10\})}{P(\{6,8,10\})} & \frac{P(\{6\})}{P(\{6,8,10\})} \end{array}\right]\\ =\left[\begin{array}{ccc} \frac{\lambda_{2}}{\lambda_{1}+\lambda_{2}+\lambda_{3}} & \frac{\lambda_{1}+\lambda_{3}}{\lambda_{1}+\lambda_{2}+\lambda_{3}} & 0\\ \frac{\lambda_{7}}{\lambda_{4}+\lambda_{5}+\lambda_{7}+\lambda_{9}+\lambda_{11}} & \frac{\lambda_{9}+\lambda_{11}}{\lambda_{4}+\lambda_{5}+\lambda_{7}+\lambda_{9}+\lambda_{11}} & \frac{\lambda_{4}+\lambda_{5}}{\lambda_{4}+\lambda_{5}+\lambda_{7}+\lambda_{9}+\lambda_{11}}\\ \frac{\lambda_{8}}{\lambda_{6}+\lambda_{8}+\lambda_{10}} & \frac{\lambda_{10}}{\lambda_{6}+\lambda_{8}+\lambda_{10}} & \frac{\lambda_{6}}{\lambda_{6}+\lambda_{8}+\lambda_{10}} \end{array}\right]. \end{array}$$

## 5. Greechie and Wright’S Twelfth Dispersionless State on the Pentagon/Pentagram/House Logic

Despite the aforementioned 11 two-valued states, there exists another dispersionless state on cyclic pastings of an odd number of contexts, namely, a state being equal to $\frac{1}{2}$ on all intertwines/bi-connections [29,42]. This state and its associated probability distribution are neither realizable by quantum nor by classical probability distributions. In this case, the conditional probabilities of any two distinct contexts $C_{i}$ and $C_{j}$, for 1≤i,j≤5 are
(10)$$\begin{array}{c} \left[P(C_{i}|C_{j})\right]\equiv \left[\begin{array}{ccc} \frac{1}{2} & 0 & \frac{1}{2}\\ 0 & 0 & 0\\ \frac{1}{2} & 0 & \frac{1}{2} \end{array}\right]. \end{array}$$

## 6. Three-Colorable Dense Points on the Sphere

There exist dense subsets of the unit sphere in three dimensions which require just three colors for associating different colors within every mutually orthogonal triple of (unit) vectors [43,44,45] forming an orthonormal basis. By identifying two of these colors with the value “0”, and the remaining color with the value “1”, one obtains a two-valued measure on this “reduced” sphere. The resulting conditional probabilities are discontinuous.

## 7. Extrema of Conditional Probabilities in Row and Doubly Stochastic Matrices

The row stochastic matrices representing conditional probabilities form a polytope in $R^{n^{2}}$ whose vertices are the $n^{n}$ matrices $T_{i}$, $i=1,\ldots ,n^{n}$, with exactly one entry 1 in each row ([46] p. 49). Therefore, a row stochastic matrix can be represented as the convex sum $\sum_{i=1}^{n^{n}}\lambda_{i}T_{i}$, with nonnegative $\lambda_{i}\geq 0$ and $\sum_{i=1}^{n^{n}}\lambda_{i}=1$.

For conditional probabilities yielding doubly stochastic matrices, such as, for instance, the quantum case, the Birkhoff theorem [26] yields more restricted linear bounds. It states that any doubly stochastic (n×n)–matrix is the convex hull of $m\leq {\left(n-1\right)}^{2}+1\leq n!$ permutation matrices. That is, if $A\equiv a_{ij}$ is a doubly stochastic matrix such that $a_{ij}\geq 0$ and $\sum_{i=1}^{n}a_{ij}=\sum_{i=1}^{n}a_{ji}=1$ for 1≤i,j≤n, then there exists a convex sum decomposition $A=\sum_{k=1}^{m\leq {\left(n-1\right)}^{2}+1\leq n!}\lambda_{k}P_{k}$ in terms of $m\leq {\left(n-1\right)}^{2}+1$ linear independent permutation matrices $P_{k}$ such that $\lambda_{k}\geq 0$ and $\sum_{k=1}^{m\leq {\left(n-1\right)}^{2}+1\leq n!}\lambda_{k}=1$.

## 8. Summary

I have attempted to sketch a generalized probability theory for multi-context configurations of observables which may or may not be embeddable into a single classical Boolean algebra. Complementarity and distinct contexts require an extension of the Kolmogorov axioms. This has been achieved by an additional axiom ascertaining that the conditional probabilities of observables in one context, given the occurrence of observables in another context, form a stochastic matrix. Various models have been discussed. In the case of doubly stochastic matrices, linear bounds have been derived from the convex hull of permutation matrices.

## Figures and Tables

**Figure 1 entropy-24-01285-f001:**
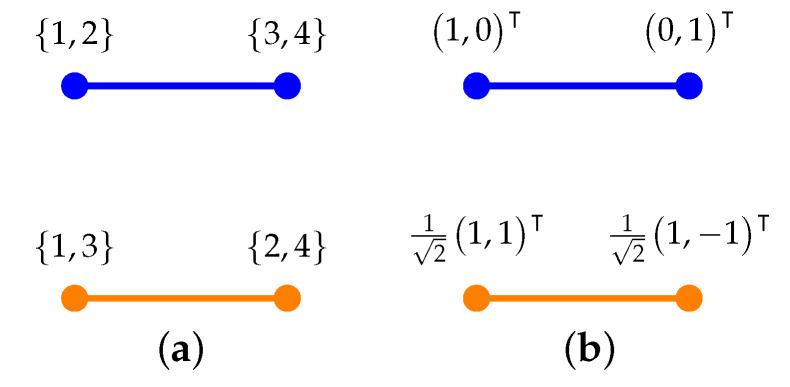
Greechie orthogonality (hyper)diagram of a logic consisting of two nonintertwining contexts: (**a**) the associated (quasi) classical partition logic representations obtained by an inverse construction using all two-valued measures therein [27]; (**b**) a faithful orthogonal representation [37] rendering a quantum *double*.

**Figure 2 entropy-24-01285-f002:**
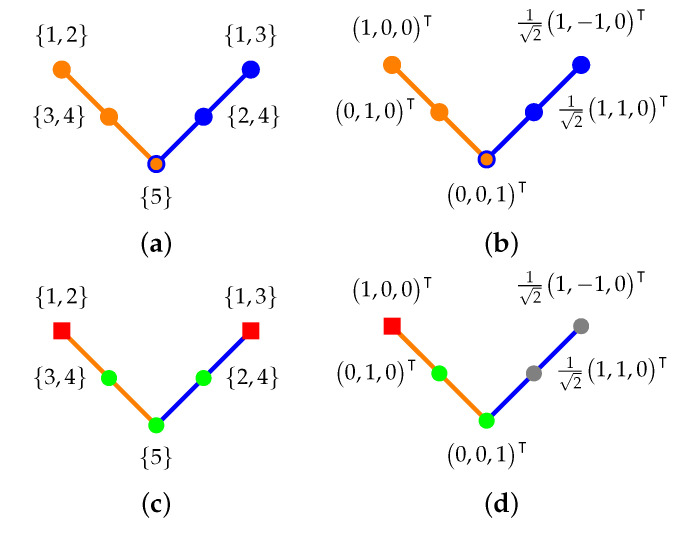
Greechie orthogonality (hyper)diagram of the $L_{12}$ “firefly” logic: (**a**) the associated (quasi)classical partition logic representation obtained through in inverse construction using all two-valued measures therein [27]; (**b**) a faithful orthogonal representation [37] rendering a quantum *double*; (**c**) the “classical” two-valued measure or truth assignment number one (of five); (**d**) a pure quantum state prepared as ${\left(\begin{array}{c} 1,0,0 \end{array}\right)}^{⊺}$. A red square and gray and green circles indicate value assignments 1, $\frac{1}{2}$ and 0, respectively.

**Figure 3 entropy-24-01285-f003:**
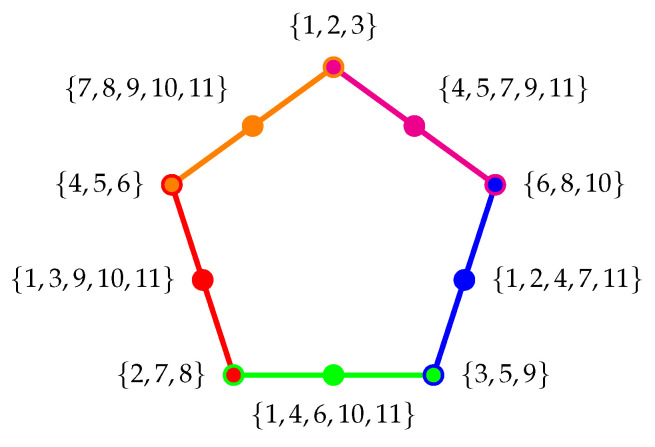
Greechie orthogonality (hyper)diagram of the pentagon/pentagram/house logic, based on eleven two-valued states.

## Data Availability

Not applicable.

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
