# Peer review of "Extending Kolmogorov’s Axioms for a Generalized Probability Theory on Collections of Contexts"

_entropy, 2022, doi:10.3390/e24091285_

Round 1

Author Response

This Referee Report brought forward conceptualizations I have not been aware of, and it pointed out a case where the manuscript needed revisions.

In particular, I have added a paragraph stating that, as expressed by the Referee,

"In terms of probabilistic language, one might interpret contexts as conditions.
Intertwining contexts might be identified with different conditionings with non-empty intersection(s)."

I also acknowledged that bi-stochasticity in quantum mechanics is valid for pure states, but need not be valid for more general states.

Therefore, I have changed the title of Section 4.1. to

"Quantum bi-stochasticity for pure states"

and added a paragraph at the end of this section with the following wording:

"It is important to emphasize that bistochasticity holds for pure states but not for more general ones.
In particular, for non-rank-one density matrices that are the product of two vectors, such as for mixed states, the above arguments do not apply."

I also mentioned this restriction to pure state at the end of Section 2.

Last, let me kindly point out that I take the impressions and general criticism mentioned by this Referee very seriously. It would be nice, in particular, for future publications, to learn more about the research alluded to by the Referee.

In any case, I would kindly like to thank the Referee for the hints and terms mentioned in the Referee Report. 

Reviewer 2 Report

This is an important paper that shows how to deal with many different contexts. This is fundamental because of the contextuality of quantum mechanics. The author extends the axioms of probability to deal with these issues. He uses row, column, and row and column stochastic matrices to get interesting results. Once the setup is done, the author goes on to show it in many different cases. While the first example is fleshed out, the rest of the examples are done very quickly.

I think this paper is without error, important for the foundations of quantum mechanics, clear and well stated. There is, however, one way that this paper can be improved. The later examples given should be elaborated on. They are written with bibliographic sources where one can read more about it. However, most people do not know these examples. Some of them can be “fleshed out” and made understandable. How much these other examples should be expanded depends on space considerations and the will of the author and the editor of the special volume.  

I found only one typo in this paper: in line 81. The C_f is wrong.  

In conclusion, I strongly urge you to publish this important and well-written paper.

Author Response

The Referee kindly observed an error in one of the expressions that I corrected accordingly. The corrected passage C_f -> C_2 is now on lines 84 and 85 and reads:

"On the other hand, if a particular element f_i \in C_2 of the second context C_2 remains fixed
and the column sum \sum_{e_j \in C_1} P(f_i|e_j) extends over all e_j \in C_1 ..."

According to the suggestions of the Referee I have added a discussion of a particular example---the "firefly logic"---and explained and motivated it in more detail

This exposition can now be found at the beginning of Section 4.2.2. Two intertwining three-atomic contexts, and extends over several new paragraphs:

"In what follows, we shall investigate a “firefly” model that has been introduced [38,
Fig. 3A.1, p. 22] to investigate a quasi-classical example of an empirical situation occurring
in quantized systems with three exclusive outcomes, formalized by three-dimensional
Hilbert space. It comprises a box with two perpendicular windows and a firefly inside.
Suppose that sometimes the firefly radiates some light, and sometimes it does not shine.
Suppose further that each one of the two perpendicular windows has a thin vertical line
drawn down the center to divide the respective window in half.

This configuration allows two types of experiments corresponding to looking through
exactly one of the two windows, respectively. Each type of experiment has three outcomes,
labeled as follows:
(i) e_1 (first type of experiment) or f_1 (second type of experiment): the light of the firefly is
in the left half of the window;
(ii) e_2 (first type of experiment) or f_2 (second type of experiment): the light of the firefly is
in the right half of the window;
(iii) e_3 (first type of experiment) and f_3 (second type of experiment): the firefly does not
shine (emit light).

The two observers at the two windows may observe any of the four combinations e_1 or f_1,
e_1 or f_2, e_2 or f_1, or e_2 or f_2. And ideally, it will always be the case that, whenever the first
observer registers no light—that is, e_3—also the second observer will register no light—that
is, f_3, and vice versa.
This firefly configuration thus gives rise to two contexts {e_1, e_2, e_3} and {f_1, f_2, f_3},
associated with the two observers, respectively. These contexts are “tied together” and
intertwine at the “no light” event or outcomes e_3 and f_3. Together this results in five
conceivable experimental outcomes for two observers, corresponding to five two-valued
measures representing these outcomes, respectively."

I hope this contributes to a better understanding of at least some of the examples involved and improves the overall comprehensibility of the manuscript.